# Antimicrobial Activity the Essential Oil from *Croton pluriglandulosus* Carn. Leaves against Microorganisms of Clinical Interest

**DOI:** 10.3390/jof9070756

**Published:** 2023-07-17

**Authors:** Rayara J. P. Carvalho, Pedro F. N. Souza, Ellen A. Malveira, Nilton A. S. Neto, Romério R. S. Silva, Gabriel L. C. Melo, Ayrles F. B. Silva, Leandro B. Lima, Cynthia C. de Albuquerque, Rafael W. Bastos, Gustavo H. Goldman, Cleverson D. T. de Freitas

**Affiliations:** 1Department of Biochemistry and Molecular Biology, Federal University of Ceará, Fortaleza 60020-181, Brazil; 2Drug Research and Development Center, Department of Physiology and Pharmacology, Federal University of Ceará, Fortaleza 60430-160, Brazil; 3Department of Fishery Engineering, Federal University of Ceará, Fortaleza 60356-000, Brazil; 4Department of Chemistry, Faculty of Exact and Natural Sciences, State University of Rio Grande do Norte, Mossoró 59650-000, Brazil; 5Department of Biological Sciences, Faculty of Exact and Natural Sciences, State University of Rio Grande do Norte, Mossoró 59650-000, Brazil; 6Department of Microbiology and Parasitology, Biosciences Center, Federal University of Rio Grande do Norte, Natal 59078-970, Brazil; 7Faculty of Pharmaceutical Sciences of Ribeirão Preto, University of São Paulo, São Paulo 14040-903, Brazil

**Keywords:** *Croton*, yeast, inhibition, chemical composition, action mechanisms

## Abstract

Multiresistant pathogens pose a serious threat to human health. The genus *Candida* is one class of human pathogenic yeasts responsible for infections affecting healthy and immunocompromised patients. In this context, plant essential oils emerged as a future natural alternative to control the diseases caused by these pathogens. Based on that, the present study aimed to evaluate the antimicrobial potential of essential oil from *C. pluriglandulosus* and understand the mechanism of action. Here, it highlighted antimicrobial activity and the mechanisms of action of the essential oil extracted from *C. pluriglandulosus* Carn.-Torres & Riina (*CpEO*) leaves on human pathogenic microorganisms in planktonic and biofilm lifestyles. In addition, for the first time, the oil composition was revealed by GC-MS analysis and the toxicity to human red blood cells (HRBC). Twenty-six chemical compounds were identified in *CpEO*, elemicin, bicyclogermacrene, caryophyllene, brevifolin, and 2,4,6-trimethoxy-styrene. Through hemolytic assay, it was shown that *CpEO* has no toxicity to human RBCs. At the concentration of 50 μg mL^−1^, *CpEO* did not show great antibacterial potential. However, promising data were found for *C. krusei* and *C. parapsilosis* inhibiting by 89.3% and 80.7% of planktonic cell growth and 83.5% and 77.9% the biofilm formation, respectively. Furthermore, the mechanisms of action *CpEO* were elucidated by fluorescence. Scanning electron microscopy revealed damage to the cell membrane and pore formation, ROS overproduction, and induction of apoptosis in candida cells. Our results reinforce the potential of *CpEO* as an effective alternative molecule of pharmaceutical interest.

## 1. Introduction

Pathogens seriously threaten human health and generate global concern due to drug resistance. The *Candida* genus holds many species, including *Candida albicans*, the most common and prevalent human pathogenic yeast [1]. However, recent studies emphasize increasing clinical cases of non-*albicans* yeasts, such as *C. glabrata*, *C. krusei*, and *C. lusitaniae*, affecting healthy and immunocompromised patients [2,3]. The infection establishment can be non-invasive or invasive, causing local and systemic manifestations [4].

The treatment of such infections is made with antifungal agents that directly interfere with the cell structure and metabolism of the fungus. However, due to the misuse, these fungi acquired resistance and became less susceptible to conventional drugs [5]. One form of resistance is the formation of biofilms, which are aggregates of microbial cells embedded in an extracellular polymer matrix that adheres to surfaces and exhibits drug resistance [6].

Plants are a source of promising alternative molecules to control infections. Among these molecules are the secondary metabolites with antimicrobial action [7,8]. Essential oils (EOs) are natural products with antifungal action. Hydrophobic and volatile substances are usually obtained through plants’ plant organs (leaves, stem, root, and fruits) [9].

The genus *Croton*, from the *Euphorbiaceae* family, adapted to the Caatinga biome, is recognized for grouping diverse species with the potential for producing essential oils and active ingredients. These plants are widely used in folk medicine as anti-inflammatory teas to reduce pain and intestinal problems [10]. In addition, some species are aromatic and produce essential oils with antifungal, insecticidal, and antimicrobial properties [11]. Among them *C. zehntneri*; *C. pulegioides Baill*; *C. jacobinensis Baill*; *C. nepetaefolius* and *C. blanchetianus* show activity against human pathogens [3,10,12,13].

Despite this large amount of data on *Croton* species, works with essential oils and their biological applications are still scarce. An example is the *C. pluriglandulosus* Carn.-Torres & Riina, for which the only information reported in the literature is about the characteristics of the plant, such as being classified as a shrub and the morphology of its leaves as ovate. It has many trichomes on the leaves, latex, and acropeculiar glands. In Brazil, it is predominantly found in regions with the caatinga biome [14]

Given the phytotherapeutic and microbial potential of species belonging to the *Croton* genus, the present work aimed to understand and determine the activity and mechanisms of action of the essential oil extracted from the leaves of *Croton pluriglandulosus* (*Cp*OE) Carn.-Torres & Riina, which is poorly studied regarding the control of microorganisms in planktonic and biofilm form. 

## 2. Materials and Methods

### 2.1. Biological Material

*C. albicans* (ATCC 10231), *C. krusei* (ATCC 6258), *C. parapsilosis* (ATCC22019), and the bacteria *B. subtilis* (ATCC 6633), *Enterobacter aerogenes* (ATCC 13048), *Escherichia coli* (ATCC 8739), *Klebsiella pneumoniae* (ATCC 10031), *Pseudomonas aeruginosa* (ATCC 25619), *Staphylococcus aureus* (ATCC 25923), and *Salmonella enterica* (ATCC 14028) were all obtained from the Laboratory of Plant Toxins, Federal University of Ceará, Brazil (Fortaleza, CE, Brazil).

### 2.2. Plant Material and Extraction of Essential Oil

The harvest of the leaves of *C. pluriglandulosus* Carn. (*CpEO*) were performed in Serra do Lima, in the city of Patu in Rio Grande do Norte (6°06′36″ S, 37°38′12″ O) state. A voucher specimen was deposited in the Dáardano de Andrade Lima Herbarium of the Federal University of Rural Semiarid (UFERSA, Angicos, RN, Brazil), with volume number MOSS 15058. The technique used for oil extraction was based on Oliveira et al. [15], with modifications established by Malveira et al. [16]. Fresh leaves of *CpEO* (approximately 630 g) were subjected to hydrodistillation in a Clevenger apparatus for 2 h. The fresh leaf biomass was weighed and placed in a 5 L volumetric flask containing 2 L of distilled water. The extracted oil was dehydrated with ether and anhydrous sodium sulfate and stored in an amber glass bottle under refrigeration until analysis.

### 2.3. Characterization of CpEO by GC/MS Analysis

Identification of the major chemical compounds of *CpEO* was performed by gas chromatography coupled to a mass spectrometer (GC/MS) analysis (Shimadzu GCMS-QP2010 SE, Kyoto, Japan), equipped with an Rtx^®^-5MS capillary column (30 m × 0.25 mm × 0.25 μm. The *CpEO* (55 μL) was placed in a vial (2 mL vial with 250 μL inserts) and injected for data acquisition. The operating conditions of the GC-MS/MS were optimized as follows: 70 eV, carrier gas (He), a flow rate of 1.7 mL.min^−1^, and a pressure of 53.5 KPa. Injector and detector interface temperatures were 25 °C and 230 °C, respectively. The oven temperature program was 100 °C for 3 min^−1^ and then 310 °C at a heating rate of 3.5 °C min^−1^ and maintained at 310 °C for 5 min^−1^. All the data acquisition was done in a run of 52 min in total. The identification of the constituents of the essential oils was investigated by comparing the mass spectra and Kovats index (IK) values with those of the research reference library. The identification was done following the method published by Adams [17].

### 2.4. Antimicrobial Activity

The anti-Candida assay with adaptations was performed using the microdilution test based on the methodology described by the Clinical and Laboratory Standards Institute (CLST, 2008, Berwyn, PA, USA). Colonies were suspended in Sabouraud Dextrose medium (twice concentrated) and diluted to (2.5 × 10^6^ CFU mL^−1^). Before the assay, the *Cp*EO solutions were prepared by dilution of concentrated oil into the desired concentrations in 5% DMSO. For antimicrobial evaluation, aliquots of 50 μL of medium and yeast cells were added to wells of 96-well plates along with 50 μL of *CpEO* (ranging from 50 to 0.008 μg mL^−1^). The antifungal drug Itraconazole (1000 μg mL^−1^ in 5% DMSO) was used as a positive control for inhibition, and DMSO 5% negative control for inhibition. The plates were incubated for 24 h at 30 °C. The yeast development was monitored using a microplate reader (Epoch, Bio-Tek Instruments, Winooski, VT, USA) at an absorbance of 600 nm. The antibacterial susceptibility assay was based on the method determined by Oliveira et al. [18], with modifications. The Mueller-Hinton Broth medium was used to grow the strains. The experimental setup followed the same principles as the abovementioned test, with only the positive control using ciprofloxacin (1000 μg mL^−1^ in 5% DMSO). For antimicrobial activity calculation, the formula was used: the OD of treated cells x 100/OD of cells treated with DMSO. The data are presented as the means of three biological replicates done individually, each with three technical replicates for each sample.

### 2.5. Antibiofilm Assay

For the evaluation of the inhibition of biofilm formation, the assay was based method described by Dias et al. [19]. The assay was based on two moments, (1) to evaluate the inhibition of biofilm formation and (2) to evaluate the ability in reduced the preformed biofilm. To evaluate the inhibition of biofilm formation, planktonic cells (2.5 × 10^6^ CFU mL ^−1^, Sabouraud Dextrose medium) were incubated with *Cp*EO for 48 h. After that, the supernatant was removed and washed three times with 0.15 M NaCl. Next, 200 μL of 100% methyl alcohol for 15 min at 37 °C was added to fix the cells. A 0.1% crystal violet solution (200 μL) was added and incubated for 30 min. Then, the wells were washed with distilled water, and 200 μL of 33% acetic acid was added for 15 min. Finally, it was read at an absorbance of 590 nm using a plate reader.

To evaluate the reduction of mature biofilm, cells (2.5 × 10^6^ CFU mL^−1^ Sabouraud Dextrose medium) were incubated for 24 h in 96-well plates to form biofilm. Then, *Cp*EO was added and incubated for 24 h to evaluate its ability to reduce biofilm mass. After that, the supernatant was removed and washed three times with 0.15 M NaCl. Next, 200 μL of 100% methyl alcohol for 15 min at 37 °C was added to fix the cells. A 0.1% crystal violet solution (200 μL) was added and incubated for 30 min. Then, the wells were washed with distilled water, and 200 μL of 33% acetic acid was added for 15 min. Finally, it was read at an absorbance of 590 nm using a plate reader.

### 2.6. Mechanisms of Action

#### 2.6.1. Cell Membrane Integrity

The concentration of 50 μg mL^−1^ was selected to be evaluated for mechanisms of action. According to Oliveira et al. [20], cell membrane integrity was analyzed with modifications. For the experiment, propidium iodide (PI) was implemented to evaluate *CpEO*-induced pore formation. First, 50 μL of *C. krusei* and *C. parapsilosis* (2.5 × 10^6^ CFU mL^−1^, in Sabouraud liquid medium) were added in the presence of 50 μL of *CpEO* (50 μg mL^−1^) and incubated for 24 h at a temperature of 30 °C. Subsequently, each treatment was washed with 100 μL of 0.15 M NaCl and centrifuged (5000× *g*, 5 min at 4 °C) three times. The samples were resuspended in 50 μL of 0.15 M NaCl and 3 μL of 1 mM propidium iodide (PI) and incubated for 30 min in the dark at 37 °C. Finally, all treated were washed twice with 0.15 M NaCl, centrifuged to remove excess PI, and observed under a fluorescence microscope (Olympus System BX 60) with an excitation wavelength of 585 nm and emission wavelength of 617 nm. Red fluorescence is indicative of membrane pore formation.

Additionally, the methodology described by Malveira et al. [16] evaluated the size of pores formed. The cells of *C. neoformans* were treated as above and incubated with 10 μM of conjugated fluorescein isothiocyanate (FITC)-Dextran with 6 kDa (Sigma Aldrich, São Paulo, SP, Brazil). After incubation for 30 min at 25 °C in the dark, the cells were washed as above and observed under a fluorescence microscope (Olympus System BX60) with an excitation wavelength of 490 nm and emission wavelength of 520 nm.

#### 2.6.2. Detection of ROS Overproduction

To evaluate *CpEO*-induced ROS generation, the method described by Dikalov and Harrison [21], with modifications, was used. All samples were prepared as previously described in Section 2.4. Subsequently, each treatment was washed with 100 μL of 0.15 M NaCl and centrifuged (5000× *g*, 5 min at 4 °C) three times. The samples were resuspended in 50 μL of 0.15 M NaCl, and 9 μL of 0.2 mM DCFH-DA (2′,7′-Dichlorofluorescein diacetate) was added and incubated for 30 min in the dark. Then, cells were washed to remove the excess of fluorophore and observed under a microscope (Olympus System BX60) using an excitation wavelength of 485 nm and an emission wavelength of 538 nm.

#### 2.6.3. Induction of Apoptosis

The induction of apoptosis in the species studied was based on using CellEvent (ThermoFisher, São Paulo, SP, Brazil), following the manufacturer’s restrictions for detecting caspase action. Samples were prepared following the pattern described in the previous tests, and 9 μL of CellEvent^®^ reagent (ThermoFisher, São Paulo, SP, Brazil) was added. Finally, washes and centrifugations were performed as before. Finally, the treatments were observed with a microscope (Olympus System BX60) using an excitation wavelength of 342 nm and an emission wavelength of 441 nm.

### 2.7. Scanning Electron Microscopy (SEM)

SEM (Billerica, MA, USA) analysis was performed to evaluate cell morphological changes based on the method described by Staniszewska et al. [22]. The cells were treated under the same conditions described in the previous sections, then fixed with 1% (*v*/*v*) glutaraldehyde in 0.15 M sodium phosphate buffer at pH 7.2 for 16 h at room temperature. This was followed by three washes with sodium phosphate buffer pH 7.2 and centrifuged (5000× *g* for 5 min at 4 °C). The samples were dehydrated with ethanol (30%, 50%, 70%, 100%, 100% (*v*/*v*) for 10 min each and centrifuged as above. Finally, 50% hexamethyldisilane (HMDS, Sigma, St. Louis, MI, USA) was diluted in ethanol, left for 10 min, followed by centrifugation, and 1000% HMDS was added. 15 μL was transferred to a coverslip to dry at room temperature. The cells were coated with a gold layer with an aluminum surface using a positron emission tomography (PET) coating machine (Emitech-Q150TES, Quorum Technologies, Lewes, UK), and the images observed on a scanning electron microscope (Quanta 450 FEG, FEI, Waltham, MA, USA).

### 2.8. Hemolytic Analysis

Hemolytic analysis of *CpEO* was evaluated with human erythrocyte blood types A, B, and O, as Oliveira et al. [20] described. Three concentrations were tested, including the one used in all the assays mentioned above. Blood was centrifuged at 5000× *g* for 5 min at 10 °C, washed with 0.15 M NaCl, and diluted to a concentration of 2.5%. Each blood type was incubated (300 μL) with the *CpEO* solution (300 μL) at concentrations of 1, 3, and 5 mg mL^−1^ for 30 min at 37 °C. DMSO-NaCl (5%) and 0.1% Triton X-100 (100%) were used as negative and positive controls for hemolysis. The samples were centrifuged (5000× *g* for 5 min at 4 °C, Eppendorf 5810 centrifuge, Hannover, Germany). Then the supernatants were collected and directed into 96-well plates, and the hemolysis (%) was calculated by an absorbance of 414 nm using a microplate reader.

### 2.9. Statistical Analysis

Three independent replicates were performed for all analyses performed. Data were subjected to the application of the ANOVA test, followed by Tukey’s method using GraphPad Prism software version 5.01 (GraphPad Software company, Santa Clara, CA, USA). A *p* < 0.05 was considered statistically significant.

## 3. Results and Discussion

### 3.1. GC-MS/MS Analysis of CpEO

The GC-MS/MS analysis identified 26 metabolites in *CpEO*. Elemicin (25.77%), bicyclogermacrene (9.37%), caryophyllene (8.99%), brevifolin (4.60%), and 2,4,6-trimethoxy-styrene (3.96%) represent the compounds in the highest amount (Supplementary Appendix A). Many metabolites identified in *CpEO* belong to the terpenoid family, classified as monoterpenes and sesquiterpenes, and present several biological activities, including antimicrobial activity [23]. Elemicin has only a few reports in the literature about its potential against human pathogens; however, Rossi et al. [24] revealed the antimicrobial activity against the bacterium *Campylobacter jejuni*. Bicyclogermacrene and 2,4,6-trimethoxy-styrene, both compounds found in *CpEO*, have antifungal, antioxidant, larvicidal, and anti-inflammatory activities [25,26,27]. Caryophyllene is a molecule with many biological activities that could be applied in health and agriculture [28,29]. Many biological properties, such as antimicrobial against fungi, bacteria, and viruses, anticancer, antioxidant, and insecticidal activities, were associated with caryophyllene [28,29]. For example, it has been reported the anti-candidal potential of caryophyllene against *C. albicans* [29].

Some compounds exhibit anti-candidal activity. One example is eucalyptol and α-terpineol (Table 1). According to the literature, eucalyptol exhibits anti-inflammatory, antioxidant, antimicrobial, and anti-candidal action [30,31]. Studies show that α-terpineol has activity against yeasts and filamentous fungi, including Candida species (PINTO et al., 2014). In addition, β-elemene, 4-terpineol, and δ-elemene were associated with antifungal, antibacterial, and anticancer activities [32,33]. Many of the components identified in *CpEO* are present in other species of the genus *Croton*, however, in different quantities [16]. This difference is related to factors such as location, climate, stage of plant development, and the organ used for the study [34,35,36].

### 3.2. Antimicrobial Activity of CpEO

The inhibitory activity of *CpEO* (50 μg mL^−1^) against fungal and bacterial strains is shown in Table 2. The *CpEO* was not effective against bacterial pathogens (Table 1). In a recent study, Malveira et al. [16] reported that the essential oil from *C. blanchetianus* is ineffective against bacteria. Nevertheless, a significant reduction in the growth of two candida species tested was observed, 89.3% for *C. krusei* and 80.7% for *C. parapsilosis* at 50 μg mL^−1^. This is a fascinating result compared to other oils [16,37,38]. For instance, Carev et al. [28] reported that the essential oil from *Centaurea triumfetii* at a concentration of 500 μg mL^−1^ inhibited 75% of the growth of *C. albicans*. This concentration is 10-fold higher than the concentration employed by *CpEO*. The inhibition of biofilm formation for the same microorganisms was 83.5% for *C. krusei* and 77.9% for *C. parapsilosis*. The search for alternatives to treating infectious diseases caused by yeasts from the *Candida* genus has a significant impact, mainly because of the antifungal resistance, which is rapidly developing due to the indiscriminate use of antifungal agents [39]. In this context, plant essential oils emerged as a promising alternative for treating these pathogens.

The inhibition of biofilm formation was an excellent achievement for *CpEO*. There were few reports on the antibiofilm potential of essential oils against fungal biofilms. Most studies are related to antibacterial biofilms, which was not found in our analysis. Biofilm is a structure of high resistance to drugs developed by pathogens to facilitate their growth and spread, as well as difficulty in the action of drugs [18]. Microbial biofilm is a complex mixture of proteins, carbohydrates, and even extracellular DNA, forming a matrix that provides a suitable environment to grow and protect cells [28]. It has been reported that biofilm development could enhance drug resistance 1000 times.

Malveira et al. [16] reported that the essential oil of *C. blanchetianus* possesses activity against *C. albicans* and *C. parapsilosis* in planktonic cells and biofilm using the same concentration tested in this study. Asdade et al. [40] showed the antifungal action of *Vitex agnus-castus* L. essential oil, rich in monoterpene and sesquiterpene compounds, against clinical *Candida* strains. Taweechaisupapong et al. [41] also obtained satisfactory results using lemongrass (*Cymbopogon flexuosus*) oil aiming at biofilm inhibition of *Candida* isolates at 0.8 and 0.4 mg mL^−1^, concentrations 16 and 8 times higher the concentration presented by *CpEO*. The essential oil from lemongrass and Pistachio also presented activity against *C. albicans* cells. The lemongrass oil presented only 30% inhibition against *C. albicans* cells at a 5 mg mL^−1^ concentration. The oil from Pistachio had activity at a 2.5 mg mL^−1^ concentration, 100 and 50 times higher than the concentration presented by *CpEO*.

A study with *C. rhamnifolioides* oil revealed activity against *Aeromonas hydrophila*, *E. coli*, *Listeria monocytogenes*, *S. Enteritidis,* and *S. aureus* [42]. Filho et al. [43] reported that the oil of the *Himatanthus obovathus* flowers was efficient against *C. krusei*; however, for *C. albicans*, the oil was ineffective. Moremi et al., 2021 pointed out the *Croton* species from Southern Africa; *C. gratissimus* Burch., *C. megalobotrys* Müll, Arg., *C. menyhartii* Gȕrke, *C. pseudopulchellus Pax*, *C. steenkampianus Gerstner,* and *C. sylvaticus Schltdl.* showed low inhibitory capacity against *Escherichia coli*. These results suggest the selectivity of essential oils against distinct species of bacteria and fungi.

Furthermore, the antimicrobial activity of *CpEO* may come from compounds such as elimicin, eucalyptol, bicyclogermacrene, α-terpineol β-elemene, 4-terpineol, and δ-elemene, already mentioned in the topic above. These compounds induce cell instability, causing membrane damage, interference with ATP synthesis, and causing extravasation of intracellular contents [44]. Additionally, it has been reported that genes such as ALS1, ALS3, and HWP1, involved in the biofilm formation process, can be silenced by the action of some plant chemical compounds [45]. This shows the effectiveness of the substances found in oils, which can act as new methods of natural control against microorganisms.

### 3.3. Mechanisms of Action

#### 3.3.1. Cell Membrane Integrity

Fluorescence microscopy was used to understand the mechanisms of action of the *CpEO*. The PI cell membrane fluorescent dye became helpful in the identification of damage to yeast cell membranes, and PI binds to the DNA structure and emits red fluorescence. This substance cannot cross the membrane of living cells because they are impermeable, so PI only moves throughout damaged membranes [46].

As expected, the control with DMSO-NaCl showed no fluorescence. However, red fluorescence indicated membrane damage in the planktonic cells of *C. krusei* and *C. parapsilosis* (Figure 1). The same damage on membranes was observed in the biofilm cells in the presence of the *CpEO* (Figure 2). Even though it seems to have more cells in the treated light field, some are not viable, as revealed by PI uptake. An et al. [47] highlight α-terpineol and terpene-4-ol as compounds that induced damage to the membranes of *Aspergillus niger*, leading to cell death. These same compounds were also identified in *CpEO*.

It is worth noting that the PI does not reveal the pore size. For this purpose, a new analysis was performed with dextran (6 kDa) in conjunction with FITC (fluorescein isothiocyanate—green fluorescence) in planktonic and biofilm cells (Figure 1 and Figure 2, right panel). The formation of pores in the membrane is revealed by the fluorescence (green), which was more robust in the biofilm, indicating a 6-kDa-sized pore in the cell membranes. The results presented by FITC-Dextran are significant. Although largely used, sometimes PI does not reveal a pore formed. That happens because PI sometimes increases the membrane permeability, which is not necessarily a pore formation. Another problem with PI is that the size of the pore is too small, allowing the cells to recover [48]. PI has a diameter size of around 0.1 nm, which is tiny and allows cells to recover or handle it in another way. In contrast, the pore formed by FITC-Dextran has a size of 1 nm being classified as a huge pore, which leads to leakage of cytoplasmic content, membrane depolarization, induction of programmed cell death, making it impossible for the cell to recover, driving cell to death. Therefore, both PI and FITC-Dextran assays provided robust results suggesting membrane pore formation [16].

It is feasible to suggest that the damage caused by *CpEO* to the cell membrane of yeasts is given the presence of components in *CpEO* that were already mentioned in previous topics. Since these compounds interfere with the enzymatic mechanism that supports the structural system of the cell wall, causing morphological damage [49]. For example, the mechanism of action of caryophyllene is focused on the cellular membrane [50]. Moo et al. [50] reported that it alters the membrane permeability as revealed by Zeta-potential measurements. This membrane permeability leads to extravasation of cellular content and death [50].

There is another point related to pore formation. The ability to induce pore formation in the membrane is a potential achievement to prospect synergism action between *CpEO* and other drugs that have cytoplasmatic targets [51,52]. This type of analysis could help to bring good results for drugs that are becoming useless. Recently, Bezerra et al. [51,52] reported the synergism of antifungal drugs with synthetic peptides. Those peptides also could induce pore formation in the membrane of *Candida* spp. Maybe these pores increase the intracellular concentration of drugs, enhancing their activity. Thinking this way, *CpEO* could also be employed with drugs, given the ability to induce pore formation.

#### 3.3.2. ROS Overproduction and Induction of Apoptosis

*CpEO* induced a slight ROS production in planktonic cells (Figure 3) and biofilm (Figure 4). As expected, DMSO-NaCl did not induce ROS in any treatment. For planktonic cells, *CpEO* induced ROS accumulation in *C. krusei* and *C. parapsilosis* (Figure 3). It is important to highlight that it is possible to see some of the particles in the light field are rests of dead cells (Figure 3 light field). However, for a biofilm lifestyle, *CpEO* was more efficient against *C. krusei* (Figure 4). In the experiment with biofilm, it is trustworthy to mention that in the light field, the number of cells in the treatment with *CpEO* is smaller than control (Figure 4—light field). Other reports of ROS overproduction in human pathogenic cells induced by essential oils [16,38,53]. Recently, it was shown that the lavender (*Lavandula angustifolia*) induced ROS accumulation in *Klebsiella pneumoniae* cells, followed by programmed cell death [38].

Essential oils’ chemical constituents are involved in ROS formation resulting from a disorder in biochemical processes [54]. Different *Cymbopogon martinii* essential oil concentrations stimulated ROS production in *A. flavus* [55]. Thymoquinone, a phytochemical compound found in essential oil, generated oxidative stress in *C. glabrata* [56]. It was recently demonstrated that the essential oil from *C. blanchetianus* leaves, rich in terpene compounds, could induce ROS in planktonic and biofilm cells of *Candida* species [16] hence corroborating with the data found in this research.

ROS are essential for maintaining cellular balance and even establishing biofilms but only at optimal levels [57]. Nonetheless, the levels of ROS must be tightly controlled; if there is an imbalance, their accumulation can cause cell apoptosis by damaging proteins, lipids, carbohydrates, and nucleic acids [58].

Keeping in mind that ROS accumulation induced by *CpEO* could lead to cell death in *C. krusei* and *C. parapsilosis,* it was performed the analysis of apoptosis in *Candida* cells. Caspase-3/7 activity was observed in all oil-treated cells, less intense in planktonic cells (Figure 5 and Figure 6). Even though it seems to have a number of cells in treated samples, cells treated with *Cp*EO, although in higher numbers, are all dead or in apoptosis-inducing event, as revealed by the green fluorescence. As expected, the control (DMSO-NaCl solution) did not induce DNA degradation, as no fluorescence was detected. Thus, *CpEO* induced DNA damage in *C. krusei* and *C. parapsilosis*. Sun et al. [59] showed that cells treated with eucalyptol and β-cyclocitral activated caspase-3 and caspase-9, proteases involved in the apoptosis process.

It is known that caspase-3 initiates apoptotic DNA fragmentation and triggers a series of reactions involved in this process [60]. It is worth noting that few studies with essential oils show this type of test in yeast, not describing the mechanism of action of the oil. A recent study revealed by proteomics that limonene induced a higher accumulation of proteins involved in DNA damage and apoptosis [61].

### 3.4. Scanning Electron Microscopy (SEM)

SEM was used to evaluate the damage caused by *CpEO* in planktonic cells of *C. krusei* and *C. parapsilosis* after all treatments. The images depict that the control cells for both species have healthy anatomy without damage (Figure 7A and Figure 8A). In contrast, the damage is visible in *CpEO*-treated cells. Among them, in cell morphology, deformations in the structure, damaged cell walls, and contorted cells (Figure 7B–F and Figure 8B–F). For both *Candida* species is clear that *CpEO* induced deadly damage. It is possible to see holes in the cell structure, indicating that some parts of the membrane and cell wall are gone.

### 3.5. Hemolytic Assay

*CpEO* did not show hemolytic activity against any human blood type tested (Table 2), even at the concentration of 3 mg mL^−1^. Thus, *CpEO* was not toxic to human red blood cells. This result reinforces the pharmacological application of *CpEO*, as it does not present toxicity risks. This is one of the strong reasons these substances can be used clinically as drugs [18]. For example, Malveira et al. [16] revealed the oil from *C. blanchetianus* was toxic to red blood cells type O.

Not harmful to human red blood cells is a critical feature of molecules that could be applied in treatment once blood is the vehicle for these molecules. Therefore, *CpEO* has excellent potential to be assessed as a source of new antimicrobial molecules.

## 4. Conclusions

Our research reveals that the essential oil from *C. pluriglandulosus* Carn.-Torres & Riina leaves is a possible alternative for controlling human pathogens that already present resistance to conventional drugs offered by the pharmaceutical industry. The data related to the mechanism of action are vital to understanding the potential application in treating infection caused by *C. parapsilosis* and *C. krusei*. Another important feature of *Cp*EO is the absence of toxicity to human red blood cells opens perspectives for the application of *Cp*EO in the development of a new treatment. Additionally, *Cp*EO showed relevance against biofilms, which is a great structure of resistance produced by yeasts. Altogether, these reveal the potential of *Cp*EO as a source of therapeutical molecules.

## Figures and Tables

**Figure 1 jof-09-00756-f001:**
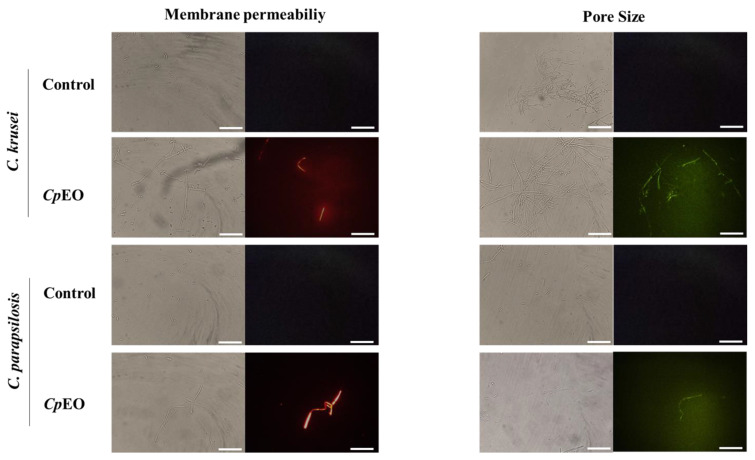
Fluorescence images showing membrane pore formation and their respective size in planktonic cells of *C. krusei* and *C. parapsilosis*. The control was DMSO-NaCl, treated with *CpEO* at 50 μg mL^−1^. Membrane pore formation was measured by propidium iodide (PI) uptake assay, and pore size used a 6 kDa dextran-FITC. White bars indicate 100 μm.

**Figure 2 jof-09-00756-f002:**
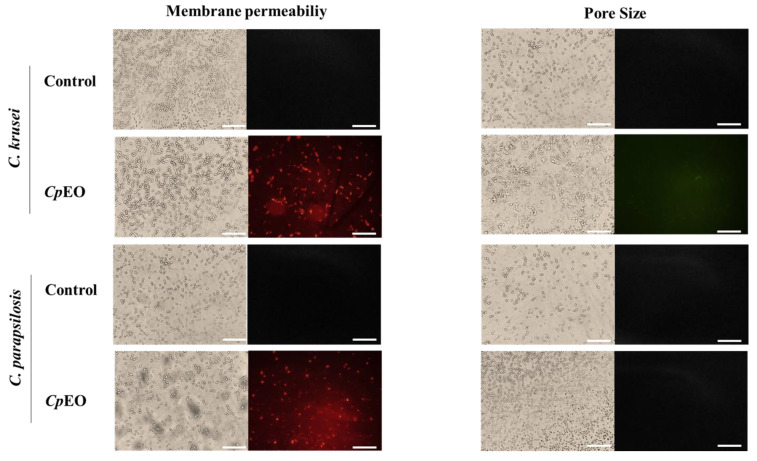
Fluorescence images showing membrane pore formation and their respective size in biofilm cells of *C. krusei* and *C. parapsilosis*. The control used was DMSO-NaCl, treated with *CpEO* at 50 μg mL^−1^. Membrane pore formation was measured by propidium iodide (PI) uptake assay, and pore size used a 6 kDa dextran-FITC. White bars indicate 100 μm.

**Figure 3 jof-09-00756-f003:**
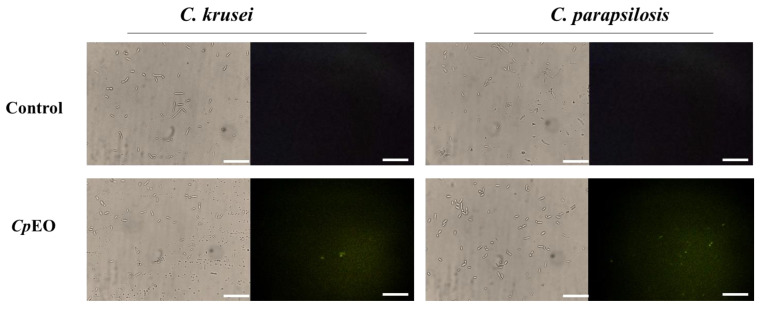
Fluorescence images showing overproduction of ROS in planktonic cells of *C. krusei* and *C. parapsilosis*. The control was DMSO-NaCl, treated with *Cp*EO at 50 μg mL^−1^. White bars indicate 100 μm.

**Figure 4 jof-09-00756-f004:**
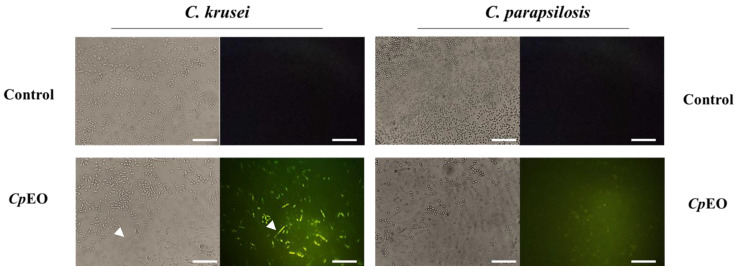
Fluorescence images showing overproduction of ROS in biofilm cells of *C. krusei* and *C. parapsilosis*. The control used was DMSO-NaCl, treated with *Cp*EO at 50 μg mL^−1^. White bars indicate 100 μm. The white arrows show the fluorescent cells in both light and fluorescence fields.

**Figure 5 jof-09-00756-f005:**
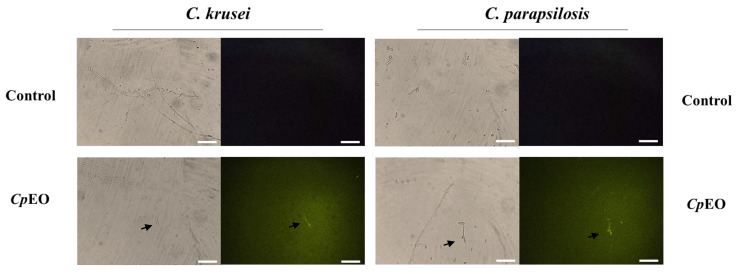
Induction of apoptosis in planktonic cells of *C. krusei* and *C. parapsilosis* using *Cp*EO at 50 μg mL^−1^ concentration. White bars indicate 100 μm. Black arrows indicate the fluorescent cells and their correspondence in the light field.

**Figure 6 jof-09-00756-f006:**
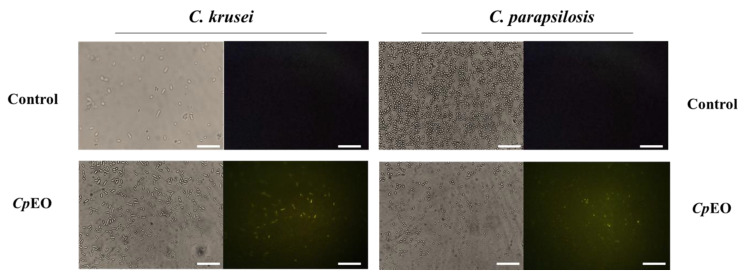
Induction of apoptosis in biofilm cells of *C. krusei* and *C. parapsilosis* using *Cp*EO at a concentration of 50 μg mL^−1^. White bars indicate 100 μm.

**Figure 7 jof-09-00756-f007:**
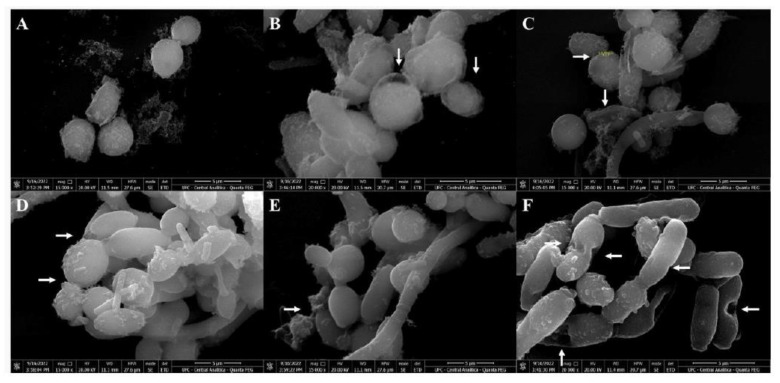
Scanning electron microscopy of *C. krusei* cells treated with *CpEO*. (**A**) Control *C. krusei* cells treated with DMSO-NaCl solution. (**B**–**F**) *C. krusei* cells treated with *CpEO*. White arrows show damage to cell structure.

**Figure 8 jof-09-00756-f008:**
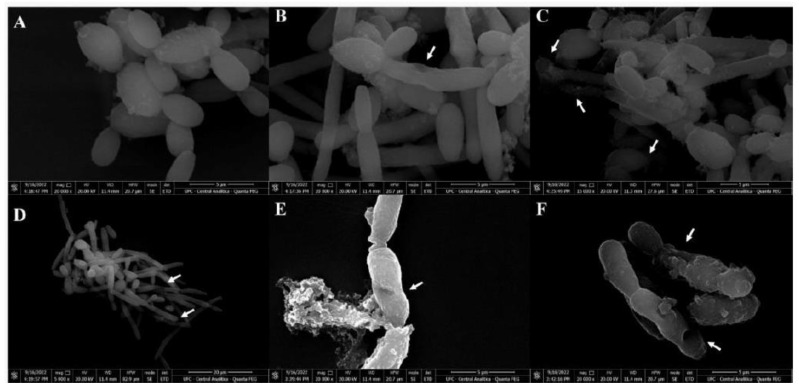
Scanning electron microscopy of *C. parapsilosis* cells treated with *CpEO*. (**A**) Control *C. parapsilosis* cells treated with DMSO-NaCl solution. (**B**–**F**) *C. parapsilosis* cells treated with *CpEO*. White arrows show damage to cell structure.

**Table 1 jof-09-00756-t001:** Antimicrobial effects of *C. pluriglandulosus* Carn. essential oil against planktonic cells and biofilm.

*Cp*EO (50 μg mL^−1^)								
**Anti-Candidal Activity**								
Microorganisms	Inhibition of Planktonic Cells Growth (%)	Significance ^a^	Inhibition of Biofilm Formation (%)	Significance ^a^	Biomass Reduction of Preformed Biofilm (%)	Significance ^a^	Itraconazole	Significance ^a^
*C. albicans*	00.00	*p* > 0.05	00.00	*p* > 0.05	00.00	*p* > 0.05	00.00	*p* > 0.05
*C. krusei*	89.3 ± 0.02	*p* < 0.05	83.5 ± 0.01	*p* < 0.05	00.00	*p* > 0.05	74.5 ± 0.05	*p* < 0.05
*C. parapsilosis*	80.70 ± 0.03	*p* < 0.05	77.9 ± 0.07	*p* < 0.05	00.00	*p* > 0.05	80.0 ± 0.03	*p* < 0.05
Antibacterial Activity							Ciprofloxacin	
*B. subtilis*	00.00	*p* > 0.05	00.00	*p* > 0.05	00.00	*p* > 0.05	67.5 ± 0.06	*p* < 0.05
*E. aerogenes*	00.00	*p* > 0.05	00.00	*p* > 0.05	00.00	*p* > 0.05	78.7 ± 0.02	*p* < 0.05
*E. coli*	00.00	*p* > 0.05	00.00	*p* > 0.05	00.00	*p* > 0.05	68.5 ± 0.007	*p* < 0.05
*K. pneumoniae*	8.76 ± 0.09	*p* > 0.05	00.00	*p* > 0.05	00.00	*p* > 0.05	72.8 ± 0.06	*p* < 0.05
*P. aeruginosa*	14.92 ± 0.04	*p* < 0.05	00.00	*p* > 0.05	00.00	*p* > 0.05	77.3 ± 0.07	*p* < 0.05
*S. aureus*	14.30 ± 0.01	*p* < 0.05	00.00	*p* > 0.05	00.00	*p* > 0.05	87.2 ± 0.07	*p* < 0.05
*S. enterica*	6.07 ± 0.01	*p* > 0.05	00.00	*p* > 0.05	00.00	*p* > 0.05	62.3 ± 0.01	*p* < 0.05

The results are presented as the mean ± SD. ^a^ indicates statistical significance or not based on *p*-value compared to control.

**Table 2 jof-09-00756-t002:** Hemolytic activity of *CpEO* Against human red Blood cells.

Blood Type	% of Hemolysis				
	0.1 % Triton	5% DMSO	*Cp*OE 1 mg mL^−1^	*Cp*OE 1.25 mg mL^−1^	*Cp*OE 3 mg mL^−1^
A	100	0	0	0	0
B	100	0	0	0	0
O	100	0	0	0	0

## Data Availability

Data are available under reasonable requirements.

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
