# Peer review of "Antimicrobial Activity the Essential Oil from Croton pluriglandulosus Carn. Leaves against Microorganisms of Clinical Interest"

_jof, 2023, doi:10.3390/jof9070756_

Round 1

Reviewer 1 Report

The manuscript entitled “Antimicrobial activity of the essential oil from Croton pluriglandulosus Carn. Leaves against microorganisms of clinical interest” by Carvalho et al. is a study of essential oil obtained from the leaves of Croton pluriglandulosus Carn, it’s chemical composition, the toxicity toward human red blood cells, and its antimicrobial activity (using different methods).

Below authors can find some of the suggestions for improving the manuscript for publication:

1. In all manuscripts evident are, I would say, typing errors. Names of the microbes should be in italics, and names of the compounds should be all with the first capital letter or not, also for the names of plant organs there is no need in using capital letters, and there should be some consistency in writing the names. Also, capital letters can be found in the middle of the sentences. Generally, authors should pay more attention on the preparation of the manuscript for submission.

2. Abbreviation for essential oil from Croton pluriglandulosus Carn is in the start (abstract) defined as CpEO, but in the manuscript one can find abbreviations such as EOCp or CbEO. In conclusion, the authors should be consistent with the abbreviations.

3. Units are also not written correctly through the manuscripts. Please revise the whole manuscript!

4. Authors tend to use the term “molecule” for essential oil. I’m not sure that “molecule” is the right term for CpEO, maybe mixture is better.

5. Method for antimicrobial activity (2.4) should be rewritten. Lines 112-113: “Itraconazole antifungal and DMSO 5% negative control was used as the positive control.” What is used as positive or negative control?? How did the authors dissolve the oil for the analysis? Also, results are presented by means of %, in the part materials and methods should be explained how the calculations have been performed.

6. Method 2.6.1. Cell membrane integrity and 2.6.2. Detection of ROS Overproduction should be explained in more detail.  It is inconclusive in the comparison with the results.

7. I would recommend extensive language revision. Many parts of the manuscript are hard to understand.

8. Conclusion should be more informative.

I would recommend extensive language revision. Many parts of the manuscript are hard to understand.

Author Response

Authors´ Response to Reviewer #1

Reviewers' comments:
Reviewer #1 - Comments to the Author

The manuscript entitled “Antimicrobial activity of the essential oil from Croton pluriglandulosus Carn. Leaves against microorganisms of clinical interest” by Carvalho et al. is a study of essential oil obtained from the leaves of Croton pluriglandulosus Carn, it’s chemical composition, the toxicity toward human red blood cells, and its antimicrobial activity (using different methods).

Below authors can find some of the suggestions for improving the manuscript for publication:

Authors’ Response

Dear Reviewer #1

We are thankful for your spending time reviewing our manuscript. Certainly, your suggestion will bring the manuscript to a higher scientific level.

Reviewer #1 – Comment #1

  1. In all manuscripts evident are, I would say, typing errors. Names of the microbes should be in italics, and names of the compounds should be all with the first capital letter or not, also for the names of plant organs there is no need in using capital letters, and there should be some consistency in writing the names. Also, capital letters can be found in the middle of the sentences. Generally, authors should pay more attention on the preparation of the manuscript for submission.

Authors’ Response #1

Thank you for that comment, and sorry for these silly mistakes. A native English revised the manuscript speak to solve all of the mistakes.

Reviewer #1 – Comment #2

  1. Abbreviation for essential oil from Croton pluriglandulosus Carn is in the start (abstract) defined as CpEO, but in the manuscript one can find abbreviations such as EOCp or CbEO. In conclusion, the authors should be consistent with the abbreviations.

Authors’ Response #2

Sorry, it was fixed.

Reviewer #1 – Comment #3

  1. Units are also not written correctly through the manuscripts. Please revise the whole manuscript!

Authors’ Response #3

Thank you. We have fixed all of them.

Reviewer #1 – Comment #4

  1. Authors tend to use the term “molecule” for essential oil. I’m not sure that “molecule” is the right term for CpEO, maybe mixture is better.

Authors’ Response #4

We have fixed it. Thank you.

Reviewer #1 – Comment #5

  1. Method for antimicrobial activity (2.4) should be rewritten. Lines 112-113: “Itraconazole antifungal and DMSO 5% negative control was used as the positive control.” What is used as positive or negative control?? How did the authors dissolve the oil for the analysis? Also, results are presented by means of %, in the part materials and methods should be explained how the calculations have been performed.

Authors’ Response #5

Sorry for this confusion. We have fixed the sentence, as you can see below:

The antifungal drug Itraconazole was used as a positive control for inhibition and DMSO 5% negative control for inhibition.

The oil was dissolved in 5% DMSO. The concentration of oil was calculated. Then, for the antimicrobial assays, the oil was dissolved in 5% DMSO.

The calculation was added in the text. Thank you.

“For antimicrobial activity calculation, was used the formula: the OD of treated cells * 100 / OD of cells treated with DMSO.”

Reviewer #1 – Comment #6

  1. Method 2.6.1. Cell membrane integrity and 2.6.2. Detection of ROS Overproduction should be explained in more detail. It is inconclusive in the comparison with the results.

Authors’ Response #6

It was fixed. Thank you.

Reviewer #1 – Comment #7

  1. I would recommend extensive language revision. Many parts of the manuscript are hard to understand.

Authors’ Response #7

Thank you for that comment, and sorry for these silly mistakes. A native English revised the manuscript speak to solve all of the mistakes.

Reviewer #1 – Comment #8

  1. Conclusion should be more informative.

Authors’ Response #8

We have improved. Thank you.

Reviewer 2 Report

Carvalho et al conducted a study where they investigated the antimicrobial activity of CpEO against microorganisms in both planktonic and biofilm states. They specifically focused on the high fungal inhibition exhibited by the essential oil and aimed to explore the potential mechanisms underlying this inhibition. The study was well-designed, and the data were clearly presented. However, there are some comments and suggestions that I would like to address.

1.       In the methods section, it is not mentioned how the solution of CpEO was prepared. It would be helpful to include this information.

2.       Regarding the inhibitory activity of CpEO, it would be beneficial to have figures showing the changes in addition to Table 2. Additionally, it is unclear how the final concentration of CpEO (50 μg/mL) was determined.

3.       One important assay that is missing in the methods section is the pore size assay.

4.       It would be valuable to know if the authors have data on the time-course changes during reactive oxygen species (ROS) production and apoptosis, as this could provide insights into the dynamics of CpEO's antimicrobial activity.

5.       The quality of Table 1 is poor, and it would be preferable to use text instead of screenshots, similar to the other tables.

6.       It would be advantageous to include the gas chromatography-mass spectrometry (GC-MS) figures of CpEO components.

7.       Figure 1 is unclear, and a scale bar is missing. Using flow cytometry could be a more quantitative way to demonstrate the changes in membrane permeability.

8.       The cell concentration in the control and CpEO groups in Figure 2 does not seem reasonable, and the background between the groups is not comparable, similar to the other fluorescent images.

9.       The quality of Figure 3 is not sufficient.

10.   In Figure 4, it appears that the light image is not aligned with the color image in the CpEO of C. Krusei.

11.   Figure 5 does not show enough cells, and the cell number is misleading between the two groups.

12.   The scale bars in Figures 7 and 8 are not clear, and the scale bar in Figure 8D is different from the others. Including transmission electron microscopy (TEM) images could provide direct evidence of membrane changes.

Moving on to minor comments:

1.       In line 35, "Induction" should be changed to "induction."

2.       In line 4, it would be helpful to clarify what is meant by "healthy patients."

3.       The unit "mg L-1" should be consistent, such as "mg·L-1" (with superscript). In line 103, "min-1" should be changed to "min," and in line 110, "mL1" should be "mL-1." Line 289 mentions itraconazole and ciprofloxacin, but concentrations are missing.

4.       In line 120, "on" is missing before "method."

5.       In line 207, the reference is not formatted correctly.

6.       In line 215, "Table 1" should be changed to "Table 2."

7.       In line 221, "C. albicans" should be italicized.

8.       In line 267, "3.3" should be on a new line.

9.       In line 289, the abbreviation "PI" has already been used before, so it is not necessary here.

10.   Regarding lines 338-348, it would be interesting to know if any component identified in the present study is related to ROS production. If so, adding a discussion on this topic would be valuable.

Author Response

Authors´ Response to Reviewer #2

Reviewer #2 Comments to the Author

Carvalho et al conducted a study where they investigated the antimicrobial activity of CpEO against microorganisms in both planktonic and biofilm states. They specifically focused on the high fungal inhibition exhibited by the essential oil and aimed to explore the potential mechanisms underlying this inhibition. The study was well-designed, and the data were clearly presented. However, there are some comments and suggestions that I would like to address.

Authors’ General Response

Dear Reviewer #2

We are thankful for you spending time to review our manuscript and for all your comments and recommendations. We have to assume that the article is much better now after all your criticisms and suggestions. Thank you.

Reviewer #2 - Comment 1

  1. In the methods section, it is not mentioned how the solution of CpEO was prepared. It would be helpful to include this information.

Authors’ comments #1

Dear reviewer #2

The information about preparation and obtention of oil is presented in the topic of oil extraction section number 2.2.

Reviewer #2 - Comment 2

  1. Regarding the inhibitory activity of CpEO, it would be beneficial to have figures showing the changes in addition to Table 2. Additionally, it is unclear how the final concentration of CpEO (50 μg/mL) was determined.

Authors’ comments #2

Dear reviewer #2,

The final concentration was obtained by a dilution of 100 µL of CpEO (at 100 µg) with 100 µL of media with cells. In that way, the final concentration of oil in the well was 50 µg in 200 µL. This a very routine calculation employed for all labs to perform antimicrobial assay.

Reviewer #2 - Comment 3

  1. One important assay that is missing in the methods section is the pore size assay.

Authors’ comments#3

Sorry for that silly mistake. We just forgotten. Now we have added in the revised version of the manuscript.

Reviewer #2 - Comment 4

  1. It would be valuable to know if the authors have data on the time-course changes during reactive oxygen species (ROS) production and apoptosis, as this could provide insights into the dynamics of CpEO's antimicrobial activity.

Authors’ comments#4

 We understand your comment. In this study we looked to see the ROS accumulation and apoptosis. We are now running a new set of experiments to understand the involvement of ROS metabolism in the activity.

Reviewer #2 - Comment 5

  1. The quality of Table 1 is poor, and it would be preferable to use text instead of screenshots, similar to the other tables.

Authors’ comments#5

We improved the quality of the tables.

Reviewer #2 - Comment 6

  1. It would be advantageous to include the gas chromatography-mass spectrometry (GC-MS) figures of CpEO components.

Authors’ comments#6

Thank for that comment. However, is not common to provide this type of information in original research because it is a huge volume of figures. We hope you understand.

Reviewer #2 - Comment 7

  1. Figure 1 is unclear, and a scale bar is missing. Using flow cytometry could be a more quantitative way to demonstrate the changes in membrane permeability.

Authors’ comments#7

We have improved figure 1. Added the scale bar in all of them. In our case, our aim was only to show the presence of red fluorescence, which indicates the pore formed. Not quantified them. We believe that those figures achieved the purpose.

Reviewer #2 - Comment 8

  1. The cell concentration in the control and CpEO groups in Figure 2 does not seem reasonable, and the background between the groups is not comparable, similar to the other fluorescent images.

Authors’ comments#8

Sorry, we don’t understand your comment. In all CpEO groups the number of cells is smaller than in other control groups. That happens because CpEO kills cells and hence reducing cell concentrations.

Reviewer #2 - Comment 9

  1. The quality of Figure 3 is not sufficient.

Authors’ comments#9

We have improved the quality. The problem is because only a few number of cells presented fluorescence.

Reviewer #2 - Comment 10

  1. In Figure 4, it appears that the light image is not aligned with the color image in the CpEO of C. Krusei.

Authors’ comments#10

It was fixed thank you.

Reviewer #2 - Comment 11

  1. Figure 5 does not show enough cells, and the cell number is misleading between the two groups.

Authors’ comments#11

The number of cells in control is clearly less than in treated groups. Additionally, we improved the misleading. It is now clear the cells in both fields.

Reviewer #2 - Comment 12

  1. The scale bars in Figures 7 and 8 are not clear, and the scale bar in Figure 8D is different from the others. Including transmission electron microscopy (TEM) images could provide direct evidence of membrane changes.

Authors’ comments#12

These imagens presented are the original from SEM scanner. In figure 8D, we had to reduce the zoom to reach all the fungal structures.

Reviewer #2 – minor comments

Moving on to minor comments:

  1. In line 35, "Induction" should be changed to "induction."

  1. In line 4, it would be helpful to clarify what is meant by "healthy patients."

  1. The unit "mg L-1" should be consistent, such as "mg·L-1" (with superscript). In line 103, "min-1" should be changed to "min," and in line 110, "mL1" should be "mL-1." Line 289 mentions itraconazole and ciprofloxacin, but concentrations are missing.

  1. In line 120, "on" is missing before "method."

  1. In line 207, the reference is not formatted correctly.

  1. In line 215, "Table 1" should be changed to "Table 2."

  1. In line 221, "C. albicans" should be italicized.

  1. In line 267, "3.3" should be on a new line.

  1. In line 289, the abbreviation "PI" has already been used before, so it is not necessary here.

  1. Regarding lines 338-348, it would be interesting to know if any component identified in the present study is related to ROS production. If so, adding a discussion on this topic would be valuable.

Authors’ comments#13

All those minor comments were fixed. Thank you.

Reviewer 3 Report

In this research article entitled " Antimicrobial activity of the essential oil from Croton pluriglandulosus Carn. Leaves against microorganisms of clinical interest" the authors explored antimicrobial activity of EO against some clinical isolates with the impact on genus Candida. Quantitatively, there were performed enough experiments and results and discussion were presented and analyzed well in most cases. Tables and figures are mainly clear and organized. The report is interesting, because nowadays we really need substitutes for antibiotics, especially due to the increasing resistance of microorganisms to currently used drugs, that's why I rate this article as beneficial. However, I mention below some points that should be considered before processing further.

-clarify the units through document, once time is describe µg and next mL, also units of inoculum: write CFU/ml or CFU.mL-1 -correct throughout the document

-Table 1: What do the values in the table mean? Are these the sizes of the inhibition zones? You have worked with 96-well plates so what is it about? The data presented the  percent growth compared to control? I don't understand this table at the same time, I am surprised that the article does not include any basic statistics, so I would like to know how SD (standard deviation – moreover this information is not mention in the Table and it is not even in the legend below the table) was calculated or are the data SEM? However, you write in the section Material and Methods that the ANOVA test (what multifactor or one-way?, followed by Tukey's method) was used. Where are the results?

-write all p-values in italics throughout the document (for example, L184)

-how did you quantify the content of the components in the essential oil? You did not use any standards or GC-MS FID and there are very few components identified. Where are the others??? Mandatory: List into the Table those components even they are not identified and add their RT !

- nowhere in the manuscript is the aim of the study specified, neither in the abstract nor at the end of the Introduction section... put it at the end of the Introduction section and at the same time briefly describe the design of the experiment

- Abstract: the genus Candida is written with a capital C L21

- Overall, the manuscript brings reduced clinical impact. It only presents in vitro data, only against ATCC strains.  I consider mandatory that clinical strains should be evaluated. In addition, multidrug-resistant strains could have also been employed.

Minor editing of English language is is required (there are some minor inconsistencies, typos and errors in the text).

Author Response

Authors´ Response to Reviewer #3

Reviewer #3 Comments to the Author

In this research article entitled “ Antimicrobial activity of the essential oil from Croton pluriglandulosus Carn. Leaves against microorganisms of clinical interest” the authors explored antimicrobial activity of EO against some clinical isolates with the impact on genus Candida. Quantitatively, there were performed enough experiments and results and discussion were presented and analyzed well in most cases. Tables and figures are mainly clear and organized. The report is interesting, because nowadays we really need substitutes for antibiotics, especially due to the increasing resistance of microorganisms to currently used drugs, that’s why I rate this article as beneficial. However, I mention below some points that should be considered before processing further.

Authors’ General Response

Dear Reviewer #3

We are thankful for you spending time to review our manuscript and for all your comments and recommendations. We have to assume that the article is much better now after all your criticisms and suggestions. Thank you.

Reviewer #3 - Comment 1

  1. -clarify the units through document, once time is describe µg and next mL, also units of inoculum: write CFU/ml or CFU.mL-1 -correct throughout the document

Authors’ comments #1

Dear Reviewer, #3

We have fixed all that information. Thank you.

Reviewer #3 - Comment 2

-Table 1: What do the values in the table mean? Are these the sizes of the inhibition zones? You have worked with 96-well plates so what is it about? The data presented the  percent growth compared to control? I don't understand this table at the same time, I am surprised that the article does not include any basic statistics, so I would like to know how SD (standard deviation – moreover this information is not mention in the Table and it is not even in the legend below the table) was calculated or are the data SEM? However, you write in the section Material and Methods that the ANOVA test (what multifactor or one-way?, followed by Tukey's method) was used. Where are the results?

Authors’ comments #2

Dear reviewer #3,

As explained in the methodology section 2.4. The assay ran out in 96-well plates by measuring the absorbance of control- and treated groups. Yes, the date is the growth compared with the control treated with 5% DMSO. Please, look back at the manuscript, in topic 2.9 we described all statistical analyses run in the manuscript. We have done one-way followed by Tukey's method.  

Reviewer #3 - Comment 3

  1. -write all p-values in italics throughout the document (for example, L184)

Authors’ comments#3

Sorry for that silly mistake. We fixed throughout the document.

Reviewer #3 - Comment 4

  1. -how did you quantify the content of the components in the essential oil? You did not use any standards or GC-MS FID and there are very few components identified. Where are the others??? Mandatory: List into the Table those components even they are not identified and add their RT !

Authors’ comments#4

 Based on your request, we provided a new table with other elements found.

Regarding quantification, what we have are the areas of the peaks in the chromatograms, it is an estimate of quantity, it is not exactly a quantification with validity method and calibration curve with standards. I recommend that you do as the reviewer recommended, include the unidentified. About quantification, it is to say that the grades were estimated according to the area of each peak.

The quantity of molecules in CpEO was determined by the Kovats index compared to the literature (Adams). Regarding the amounts, no compound quantification procedure was carried out, those percentages are values of the peak areas in the chromatograms that serve as an estimate of the amounts

Reviewer #3 - Comment 5

  1. - nowhere in the manuscript is the aim of the study specified, neither in the abstract nor at the end of the Introduction section... put it at the end of the Introduction section and at the same time briefly describe the design of the experiment

Authors’ comments#5

The information was added to the new version of the manuscript. Thank you.

Reviewer #3 - Comment 6

  1. - Abstract: the genus Candida is written with a capital C L21

Authors’ comments#6

It was fixed. Thank you.

Reviewer #3 - Comment 7

  1. - Overall, the manuscript brings reduced clinical impact. It only presents in vitro data, only against ATCC strains. I consider mandatory that clinical strains should be evaluated. In addition, multidrug-resistant strains could have also been employed.

Authors’ comments#7

We understand your concern. But we first understand the potential of CpEO against ATCC strains and evaluate the mechanism of actions. Now, in a second part in collaboration we will try the activity against clinical strains. We don´t have a lab qualified to work with such pathogens. So, we first study ATCC and them move to resistant strains.

Round 2

Reviewer 1 Report

Dear authors, some progress compared to the first version of the manuscript is evident, but again, the same lack can be detected. I would recommend you pay attention to detail and take the time to improve the manuscript. Below you can find some specific comments. 

Line 29: “Bicyclogermacrene” first letter should not be capitalized.

Line 56: “(Leaves, Stem, Root and fruits)”, again, consistency is needed. All first letters should be capitalized or not.

Line 272: „C. albicans” should be italics-C. albicans

In part 2. Material and method, subheading 2.3. Characterization of CpEO by GC/MS Analysis, it’s lacking the time of analysis performed. Also, if the authors have compared retention indices with Adams, the reference is lacking too.

In part 2. Material and method, subheading 2.4. Antimicrobial activity, the authors should add the part with an explanation of preparing (dissolving) essential oil for analysis. Also, did the authors use as negative control 5% DMSO and positive control antifungal itraconazole and ciprofloxacin dissolved in medium, water, or something else? That should be explained also.

In the Material and method part, subheading 2.5. Antibiofilm assay, the first two sentences are not clear enough.

In the Material and method part, subheading 2.6. Mechanisms of Action, 2.6.1. Cell membrane integrity, the authors mention that they have used minimum inhibitory concentration for testing, but in the manuscript that concentration is not mentioned. Only the highest applied concentration was used for all evaluations, even for the discussion of the results obtained from the antimicrobial testing. Maybe this part can be better explained.

Minimum inhibitory concentration is defined as the lowest concentration that inhibits the visible growth of a microorganism. In the case of this study, a concentration of 50μg/mL inhibits 89.3% of C. krusei and 80.7% of C.parapsilosis

In part 3. Results and Discussion, subheading 3.1. GC-MS/MS analysis CpEO by- first the title should be rewritten; I suppose it should be GC-MS/MS analysis of CpEO. In Table 1 third row, there should be compounds, not composto. Also, dots in the names of compounds should be exchanged by commas. In the name of the p-cymene “p” should be italic. And again, all the first letters of the compound's names should be uniform.  

In part 3. Results and Discussion, subheading 3.2. Antimicrobial Activity of CpEO, Line 274: the word "respectively" is redundant here, since after % there is the name of inhibited bacteria.

Table 2: the units of the concentration in the Table are not written appropriately, and an explanation of the results presented, below the table, is missing (are the results presented as mean ± SD, or mean ± SV....)

Line 310: “Elimicin, Eucalyptol, Bicyclogermacrene, α-terpineol β-elemene, 4-terpineol and….” again, authors should pay attention to the consistency of writing the names of compounds.

Again, units are not consistent throughout the manuscript. As an example, please see Lines 265, 299, 438, and Table 3. Units of concentrations should be written as μg/mL, or μg.mL-1

Conclusion:

The sentence added to the conclusion part should be rewritten (Lines 455-458). It is inconclusive this way.  

Minor editing of English language required.

Author Response

Authors´ Response to Reviewer #1

Reviewers' comments:
Reviewer #1 - Comments to the Author

Dear authors, some progress compared to the first version of the manuscript is evident, but again, the same lack can be detected. I would recommend you pay attention to detail and take the time to improve the manuscript. Below you can find some specific comments. 

Authors’ Response

Dear Reviewer #1

We are thankful for your spending time reviewing our manuscript. Certainly, your suggestion will bring the manuscript to a higher scientific level.

We will do our best to reach your expectations.

Reviewer #1 – Comment #1

Line 29: “Bicyclogermacrene” first letter should not be capitalized.

Authors’ Response #1

Sorry for these silly mistakes. It was fixed.

Reviewer #1 – Comment #2

Line 56: “(Leaves, Stem, Root and fruits)”, again, consistency is needed. All first letters should be capitalized or not.

Authors’ Response #2

Sorry, it was fixed.

Reviewer #1 – Comment #3

In part 2. Material and method, subheading 2.3. Characterization of CpEO by GC/MS Analysis, it’s lacking the time of analysis performed. Also, if the authors have compared retention indices with Adams, the reference is lacking too.

Authors’ Response #3

Thank you. We have added all the information.

Reviewer #1 – Comment #4

In part 2. Material and method, subheading 2.4. Antimicrobial activity, the authors should add the part with an explanation of preparing (dissolving) essential oil for analysis. Also, did the authors use as negative control 5% DMSO and positive control antifungal itraconazole and ciprofloxacin dissolved in medium, water, or something else? That should be explained also.

Authors’ Response #4

We have fixed it. Thank you.

Reviewer #1 – Comment #5

In the Material and method part, subheading 2.5. Antibiofilm assay, the first two sentences are not clear enough.

Authors’ Response #5

Sorry for this confusion. We have fixed the sentence.

Reviewer #1 – Comment #6

In the Material and method part, subheading 2.6. Mechanisms of Action, 2.6.1. Cell membrane integrity, the authors mention that they have used minimum inhibitory concentration for testing, but in the manuscript that concentration is not mentioned. Only the highest applied concentration was used for all evaluations, even for the discussion of the results obtained from the antimicrobial testing. Maybe this part can be better explained.

Minimum inhibitory concentration is defined as the lowest concentration that inhibits the visible growth of a microorganism. In the case of this study, a concentration of 50μg/mL inhibits 89.3% of C. krusei and 80.7% of C.parapsilosis

Authors’ Response #6

We fixed that. Sorry!

We indeed used the concentration of 50 µg to perform the analysis. Now it is clear in the methodology.

Reviewer #1 – Comment #7

In part 3. Results and Discussion, subheading 3.1. GC-MS/MS analysis CpEO by- first the title should be rewritten; I suppose it should be GC-MS/MS analysis of CpEO. In Table 1 third row, there should be compounds, not composto. Also, dots in the names of compounds should be exchanged by commas. In the name of the p-cymene “p” should be italic. And again, all the first letters of the compound's names should be uniform.  

Authors’ Response #7

You right. The correct is GC-MS/MS analysis of CpEO. We fixed it.

Sorry for those mistakes. They were all fixed.

Reviewer #1 – Comment #8

In part 3. Results and Discussion, subheading 3.2. Antimicrobial Activity of CpEO, Line 274: the word "respectively" is redundant here, since after % there is the name of inhibited bacteria.

Authors’ Response #8

We fixed it. Thank you.

Reviewer #1 – Comment #9

Table 2: the units of the concentration in the Table are not written appropriately, and an explanation of the results presented, below the table, is missing (are the results presented as mean ± SD, or mean ± SV....)

Authors’ Response #8

We fixed it. Thank you.

Reviewer #1 – Comment #10

Line 310: “Elimicin, Eucalyptol, Bicyclogermacrene, α-terpineol β-elemene, 4-terpineol and….” again, authors should pay attention to the consistency of writing the names of compounds.

Authors’ Response #8

We fixed it. Thank you.

Reviewer #1 – Comment #11

Again, units are not consistent throughout the manuscript. As an example, please see Lines 265, 299, 438, and Table 3. Units of concentrations should be written as μg/mL, or μg.mL-1

Authors’ Response #8

We fixed it. Thank you.

Reviewer #1 – Comment #11

Conclusion:

The sentence added to the conclusion part should be rewritten (Lines 455-458). It is inconclusive this way. 

Authors’ Response #8

We rewrite, again, the conclusion.

Reviewer 2 Report

Dear authors,

Thank you for the replies. I still have some questions.

I requested to add the preparation of the CpEO solution, not the oil of CpEO in section 2.2 which describes the extraction of the oil from CpEO.

I would like to see the data used to determine the final concentration of CpEO, not the calculation. The corresponding figures of various concentrations could better show the inhibitory effect of CpEO.

GC-MS analysis figures of CpEO components could be included in the supplementary if it’s not appropriate for the main text.

In Figure 2 right panel of CpEO, the cell concentration seems higher than the control to me. Similar to the right panel of Figure 3 and the left panel of Figure 6.

In Figure 4, need to confirm the same field is for light and fluorescent images in the CpEO of C. Krusei?

Author Response

Authors´ Response to Reviewer #2

Reviewer #2 Comments to the Author

Dear authors,

Thank you for the replies. I still have some questions.

Authors’ General Response

Dear Reviewer #2

We are thankful for you spending time to review our manuscript and for all your comments and recommendations. We must assume that the article is much better now after all your criticisms and suggestions. Thank you.

Reviewer #2 - Comment 1

  1. I requested to add the preparation of the CpEO solution, not the oil of CpEO in section 2.2 which describes the extraction of the oil from CpEO.

Authors’ comments #1

It was added in the 2.3 section. Thank you.

Reviewer #2 - Comment 2

  1. I would like to see the data used to determine the final concentration of CpEO, not the calculation. The corresponding figures of various concentrations could better show the inhibitory effect of CpEO.

Authors’ comments #2

Dear reviewer #2,

There was no data about the dilutions. As we explained in the last answer, in the very first well, the final concentration of oil in the well was 50 µg in 200 µL. Then, based on that was done a two-fold serial dilution. Again, this is a very routine calculation employed for all labs to perform antimicrobial assay.

We decided to show only the effective concentration of CpEO.

Reviewer #2 - Comment 3

  1. GC-MS analysis figures of CpEO components could be included in the supplementary if it’s not appropriate for the main text.

Authors’ comments#3

Thank you for that comment. We moved to supplementary data.

Reviewer #2 - Comment 4

  1. In Figure 2 right panel of CpEO, the cell concentration seems higher than the control to me. Similar to the right panel of Figure 3 and the left panel of Figure 6.

Authors’ comments#4

 Dear reviewer #2,

First, these are images in Fig. 2. of biofilm assays. Even though it seems to have more cells in treated light field you can that all of them are not viable, as revealed by PI uptake. Additionally, crystal violet assay was used.

In the case of Fig. 3, some of the particles in the light field are rests of dead cells.

Regarding Fig. 6, it is the same as what happens in figure 2. But it is clear that cells treated with CpEO, although in higher numbers, are all dead or in apoptosis-inducing event as revealed by the green fluorescence.

Reviewer #2 - Comment 5

  1. In Figure 4, need to confirm the same field is for light and fluorescent images in the CpEO of C. Krusei?

Authors’ comments#5

Dear reviewer #2,

We have checked that. It is the same field. Please look at the cell presented by white arrowed read.

Reviewer 3 Report

The authors tried to improve the article, which they partly succeeded in, but this manuscript still has flaws that need to be corrected. Some are repeated as if the authors did not correct what was written before.   -Write all Latin names of microorganisms in italics (f.eg. L149, L258 etc ) in the entire document. - Table 2 is still not clear to me whether it is behind the resulting values SD or SEM! Put it in the table! Also, I still don't see any statistics, no data comparison... Statistics are clearly desirable in this article, as the authors also state that they worked with ANOVA! Please complete the statistics! - p-value should be written in italics throughout the document (e.g. L201)   Thanks to the authors for explaining some things related to clinical isolates. I recommend the authors to supplement and expand the introduction with more information and also the discussion.  

Minor editing of English language is still required.

Author Response

Authors´ Response to Reviewer #3

Reviewer #3 Comments to the Author

The authors tried to improve the article, which they partly succeeded in, but this manuscript still has flaws that need to be corrected. Some are repeated as if the authors did not correct what was written before.   -Write all Latin names of microorganisms in italics (f.eg. L149, L258 etc ) in the entire document. - Table 2 is still not clear to me whether it is behind the resulting values SD or SEM! Put it in the table! Also, I still don't see any statistics, no data comparison... Statistics are clearly desirable in this article, as the authors also state that they worked with ANOVA! Please complete the statistics! - p-value should be written in italics throughout the document (e.g. L201)   Thanks to the authors for explaining some things related to clinical isolates. I recommend the authors to supplement and expand the introduction with more information and also the discussion.  

Authors’ General Response

Dear Reviewer,

The italic names were fixed.

The information about SD was added at the end of the table.

The statistical description was improved.

Round 3

Reviewer 1 Report

The authors have significantly improved the manuscript entitled “Antimicrobial activity of the essential oil from Croton pluriglandulosus Carn. Leaves against microorganisms of clinical interest”. Mainly all the required changes have been introduced. But prior to the publication, I would just recommend a few more:

Line 255, 284, 319, 324, 424, Table 2: units are not presented correctly.

Lines 336-339: I recommend adding a reference for the statement. “In contrast, the pore formed by FITC-Dextran has a size of 1 nm being classified as a huge pore, which leads to leakage of cytoplasmic content, membrane depolarization, induction of programmed cell death, making it impossible for the cell to recover, and driving cell to death.”

Author Response

Authors´ Response to Reviewer #1

Reviewers' comments:
Reviewer #1 - Comments to the Author

The authors have significantly improved the manuscript entitled “Antimicrobial activity of the essential oil from Croton pluriglandulosus Carn. Leaves against microorganisms of clinical interest”. Mainly all the required changes have been introduced. But prior to the publication, I would just recommend a few more:

Line 255, 284, 319, 324, 424, Table 2: units are not presented correctly.

Lines 336-339: I recommend adding a reference for the statement. “In contrast, the pore formed by FITC-Dextran has a size of 1 nm being classified as a huge pore, which leads to leakage of cytoplasmic content, membrane depolarization, induction of programmed cell death, making it impossible for the cell to recover, and driving cell to death.”

Authors’ Response

Dear Reviewer #1

We are thankful for your spending time reviewing our manuscript. Certainly, your suggestion will bring the manuscript to a higher scientific level. We will do our best to reach your expectations.

All your comments were addressed. We fixed the units and added the reference.

Reviewer 2 Report

Thanks for the response. 

As for comment 4, if the cells in the white field of CpEO are dead as you explained, then why the control group has much fewer cells in the white field? I would recommend adding your answer to the main text to avoid confusion if you insist on your opinion.

Author Response

Authors´ Response to Reviewer #2

Reviewer #2 Comments to the Author

Thanks for the response. 

As for comment 4, if the cells in the white field of CpEO are dead as you explained, then why the control group has much fewer cells in the white field? I would recommend adding your answer to the main text to avoid confusion if you insist on your opinion.

Authors’ General Response

Dear Reviewer #2

Thank you for this comment. We have added all the information in the new version of the manuscript.

Reviewer 3 Report

Dear authors,

The manuscript has been improved, but there are still errors in it. For example, p value must be in italics! In addition, the authors wrote that they supplemented everything and improved the statistics, but I do not see anything in the article supplemented with statistics, only the standard deviation. So why did the authors use ANOVA and Tukey HSD test for data processing??? In the last revision I recommended adding new and some longer information to the introduction and also to the discussion, this is also not added.

Author Response

Authors´ Response to Reviewer #3

Reviewer #3 Comments to the Author

Dear authors,

The manuscript has been improved, but there are still errors in it. For example, p value must be in italics! In addition, the authors wrote that they supplemented everything and improved the statistics, but I do not see anything in the article supplemented with statistics, only the standard deviation. So why did the authors use ANOVA and Tukey HSD test for data processing??? In the last revision I recommended adding new and some longer information to the introduction and also to the discussion, this is also not added.

Authors’ General Response

Dear Reviewer,

Dear reviewer #3

We have put all p value in italics.

Regarding the statistics, this is method we used in many published papers. We used this method because is the best method for our data.

Regarding the introduction and discussion, we believe the information and size are ok for the journal, and the information is enough for readers. None of the other two reviewers have concerns about the introduction or discussion.

Round 4

Reviewer 1 Report

The authors have introduced all suggested changes.

Author Response

Reviewer #1 comment

The authors have introduced all suggested changes.

Author response

Thank you.

Reviewer 2 Report

I appreciate your reply.

I still have reservations regarding the quality of the figures. Perhaps that assessment falls within the jurisdiction of the journal's editorial team.

Author Response

Reviewer #2

I appreciate your reply.

I still have reservations regarding the quality of the figures. Perhaps that assessment falls within the jurisdiction of the journal's editorial team.

Author´s response

Dear reviewer #2, 

These are the figures we got. They are very clear to show the damage. 

We have many of these figures published everywhere. including this same journal. https://doi.org/10.3390/jof8111147

Based on that, we ask you to reconsider.

please see the papers. 

 https://doi.org/10.3390/antibiotics11121753

https://doi.org/10.3390/antibiotics11050553

Reviewer 3 Report

Dear editor and authors, I see that you have corrected and edited the article, but I do not see any request for statistics, only basic, that is, SD and conversion of turbidity to growth percentages... When I suggested to completed the statistics, I had this in mind (one way Anova - Tukey HSD test, which you also used chapter 2.9...I wanted you to use this: an application of multiple comparisons to determine which means are significantly different from which others This test shows the estimated difference between each pair of means and shows that the pairs show statistically significant differences at the 95 level .0% confidence level, all homogeneous groups are identified by columns of X. Within each column, the levels containing X form a group of means within which there are no statistically significant differences. The method currently used to distinguish between means is actually Tukey's significant difference (HSD) procedure. With this method, there is a 5.0% risk that one or more pairs will be significantly different when their true difference is 0. However, the authors did not supplement this. However, if I wanted you to supplement the information in introduction, I have my own reasons for that.. I don't know what the other reviewers wrote, I had the opinion that the introduction should be supplemented. In my opinion, reviews are done in order to reveal as many errors as possible in documents, otherwise, the editor would use 1 reviewer for the review and not 3 or 4. However, you must realize that you are publishing in a very prestigious and well-reviewed journal, where such errors should not occur, and certainly not after 3 revisions of the same document! Good luck with the article.

Author Response

Reviewer #3 comments

Dear editor and authors, I see that you have corrected and edited the article, but I do not see any request for statistics, only basic, that is, SD and conversion of turbidity to growth percentages... When I suggested to completed the statistics, I had this in mind (one way Anova - Tukey HSD test, which you also used chapter 2.9...I wanted you to use this: an application of multiple comparisons to determine which means are significantly different from which others This test shows the estimated difference between each pair of means and shows that the pairs show statistically significant differences at the 95 level .0% confidence level, all homogeneous groups are identified by columns of X. Within each column, the levels containing X form a group of means within which there are no statistically significant differences. The method currently used to distinguish between means is actually Tukey's significant difference (HSD) procedure. With this method, there is a 5.0% risk that one or more pairs will be significantly different when their true difference is 0. However, the authors did not supplement this. However, if I wanted you to supplement the information in introduction, I have my own reasons for that.. I don't know what the other reviewers wrote, I had the opinion that the introduction should be supplemented. In my opinion, reviews are done in order to reveal as many errors as possible in documents, otherwise, the editor would use 1 reviewer for the review and not 3 or 4. However, you must realize that you are publishing in a very prestigious and well-reviewed journal, where such errors should not occur, and certainly not after 3 revisions of the same document! Good luck with the article.

Author´s Response

Dear #3 

Please don´t get me wrong. But I do not understand your real concern about our data. we have published dozens of studies using this same statistical analysis that was revised by many reviewers worldwide. Nobody has even claimed our statistical analysis. However, we did what you asked and added a new column in the table with the significance and p-value. For sure, it is important to mention that all data have risks. That´s why we provided statistical analysis to support the data.  

Regarding your suggestion on introduction, it is a suggestion. We, as authors, should have the autonomy to decide or not follow it. you said you had your own reason, which was not our reason. We are the authors of the manuscript. Therefore, we kindly ask you to reconsider!